# Fast Low-Precision Computer-Generated Holography on GPU

**David Blinder** [1,2,]* and **Peter Schelkens** [1,2]

1   Department of Electronics and Informatics (ETRO), Vrije Universiteit Brussel (VUB), B-1050 Brussels, Belgium; peter.schelkens@vub.be
2   imec, Kapeldreef 75, B-3001 Leuven, Belgium
*   Correspondence: david.blinder@vub.be; Tel.: +32-2629-1694

**Abstract:** Computer-generated holography (CGH) is a notoriously difficult computation problem, simulating numerical diffraction, where every scene point can affect every hologram pixel. To tackle this challenge, specialized software instructions and hardware solutions are developed to significantly reduce calculation time and power consumption. In this work, we propose a novel algorithm for high-performance point-based CGH, leveraging fixed-point integer representations, the separability of the Fresnel transform and using new look-up table free cosine representation. We report up to a 3-fold speed up over an optimized floating-point GPU implementation, as well as a 15 dB increase in quality over a state-of-the-art FPGA-based fixed-point integer solution.

**Keywords:** digital holography; computer-generated holography; massively parallel computing; 3D displays; computer graphics

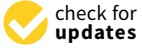



## 1. Introduction

Electro-holographic displays are a promising technology for immersive 3D displays, thanks to their ability to fully reproduce the wavefield of light, thereby accounting for all human visual cues [1,2]: continuous motion parallax, and no accommodation-vergence conflict, all while supporting accurate shading and occlusion cues. One major challenge for realizing those display systems is computational, because holograms are computed by modeling numerical diffraction: every point in the virtual 3D scene creates spherical waves that can affect every hologram pixel. This many-to-many mapping, combined with the large needed resolutions and low frame rates required for driving holographic video displays, makes the use of efficient algorithms and hardware solutions a necessity.

Nowadays, many different CGH algorithms exist that are specialized for different kinds of holography applications [3]. The best choice depends on the hologram resolution, viewing angle, display type, power and visual quality requirements. Examples of CGH algorithms are polygonal CGH [4–6], layer-based CGH [7–9], and holographic stereograms [10,11]. These can be combined with acceleration structures, such as wavefront recording planes [12], sparse bases [13,14] and deep neural networks [15].

One of the most widely used algorithms are point-based CGH [16,17]. These decompose 3D objects into a large collection of infinitesimal self-luminous points, whose point-spread functions (PSF) are calculated and superimposed to obtain the final hologram. Thanks to the relatively simple mathematical expression of PSFs and the inherent parallelizability of the linear superpositions, they are a good candidate for massively parallel implementations on application-specific integrated circuits (ASIC) [18], field-programmable gate arrays (FPGA) [19–21] and graphics processing units (GPU) [16,22].

Our goal is to optimize point-based CGH algorithms by reducing both the memory and computational requirements for calculating PSFs, targeting specialized GPU, FPGA or ASIC implementations. This is achieved by reducing the bit-width of intermediate results and approximating mathematical operators, all while preserving sufficient visual quality. Our main contributions are the following:

- We propose a generic framework for the efficient calculation of separable point-based CGH, using fixed-precision integer representations.
- We propose a more accurate, look-up-table-free approximation of (co)sine functions resulting in a 15 dB quality gain over the triangular wave implementation [19].
- A high-performance CUDA implementation for GPU is detailed with well-chosen assembly instructions, resulting in up to 3-fold speed ups over an optimized floating-point CGH implementation.

The paper is structured as follows: in Section 2, we derive the expressions optimized for low-memory and low-precision representations and detail the steps of the general algorithm. Then, we expound on the optimized CUDA implementations, explaining how these algorithms can be translated into the appropriate instructions and data structures in Section 3. This is followed by Section 4 in which we render different holograms to quantify visual quality and to measure the reductions in calculation times. Finally, we discuss the results and their interpretation in Section 5 and conclude in Section 6.

## 2. Methodology

The Fresnel model for the complex amplitude of a single PSF created by a point emitter located at coordinates $(\delta, \epsilon, \zeta)$ is given by the following expression:

$$P(x,y) = a \cdot \exp\left(\frac{\pi i}{\lambda \zeta}\left[(x-\delta)^2 + (y-\epsilon)^2\right]\right) \tag{1}$$

where $\lambda$ is the wavelength and $a$ is the point amplitude. To obtain the complex-valued hologram $H(x,y)$, we must sum over all $Q$ points in all pixels of $H$. This gives us the following:

$$H(x,y) = \sum_{j=1}^{Q} P_j(x,y) = \sum_{j=1}^{Q} a_j \cdot \exp\left(\frac{\pi i}{\lambda \zeta_j}\left[(x-\delta_j)^2 + (y-\epsilon_j)^2\right]\right). \tag{2}$$

In conventional point-based CGH, this expression is evaluated independently for all pixels, typically computed using floating-point arithmetic. We propose to significantly accelerate this by combining two features:

1. Leveraging the separability of every $P(x,y)$ along its spatial dimensions $x$ and $y$;
2. Using low-precision fixed-point integer approximations for the various mathematical operations of (2).

These are described in the remainder of this section.

### 2.1. Packed Separable PSF Phases

The phase of the PSF $P(x,y)$ is given by the parabola as follows:

$$\phi(x,y) = \angle P(x,y) = \frac{\pi}{\lambda \zeta}\left[(x-\delta)^2 + (y-\epsilon)^2\right] = \frac{\pi}{\lambda \zeta}(x-\delta)^2 + \frac{\pi}{\lambda \zeta}(y-\epsilon)^2 \tag{3}$$

i.e., scaled by a constant inversely proportional to the axial distance of the point emitter. Note that the phase $\phi$ is, therefore, separable in $\phi_x$ and $\phi_y$:

$$\phi_x(x) = \frac{\pi}{\lambda \zeta}(x-\delta)^2; \quad \phi_y(y) = \frac{\pi}{\lambda \zeta}(y-\epsilon)^2; \quad \phi(x,y) = \phi_x(x) + \phi_y(y). \tag{4}$$

This is leveraged to minimize redundant operations when calculating and packing the bits encoding the $\phi_x$ and $\phi_y$ values.

When using fixed-point integers, one should carefully select the units so as to optimize the number of required bits for encoding numbers in a given application, balancing precision and computational performance constraints. For the conversion to $n$-bit (fixed-point) integer representations, we want to encode units as integer multiples of $\pi \cdot 2^{1-n}, n \in \mathbb{N}_0$; we

make this choice since the exponential of purely imaginary functions is periodic (consisting of sines/cosines), so we can ignore any overflow when adding phase values together, which is some multiple of $2\pi$.

The precision of the quantization is denoted as $s$, which should be no larger than the hologram pixel pitch $p$ in order to ensure numerically distinguishable signal changes across subsequent pixels; otherwise, the coordinates for two neighboring pixels may be quantized (e.g. rounded) to the same value, thereby resulting in erroneous identical computed PSF amplitudes. One may want to choose $s < p$ if finer lateral quantizations of the point cloud would be desirable, e.g., for sub-pixel lateral resolution of point coordinates for very detailed objects.

These precision considerations must also be taken to determine the axial quantization of the factor $\frac{\pi}{\lambda\zeta}$, which we denote as $c$, i.e., the depth resolution of the point cloud. It should also be proportional to some $\pi \cdot 2^{1-m}, m \in \mathbb{N}_0$ so to facilitate integer multiplication, but not necessarily the same as the precision chosen for $s$. We chose the same precision for this parabolic phase scaling factor as that for $s$ because of GPU register size constraints, giving us the following integer:

$$c = \left\lfloor \frac{\pi}{\lambda\zeta} \frac{s^2 2^n}{2\pi} \right\rceil = \left\lfloor \frac{s^2}{2\lambda\zeta} \right\rceil \tag{5}$$

where $\lfloor \cdot \rceil$ is the rounding operator.

This process is summarized for $\phi_x$ in Figure 1; it is identical for $\phi_y$, except for the other chosen coordinate difference $(y - \epsilon)$. The final multiplication result can have a different bit-width $k < n$ than the intermediate integers to significantly reduce the multiplier design in, for example, a customized FPGA implementation. Because typical GPUs are more constrained, namely, memory is addressed byte-wise and registers are 32-bit for most modern GPU models, we chose $k = 8$ and $n = 32$.

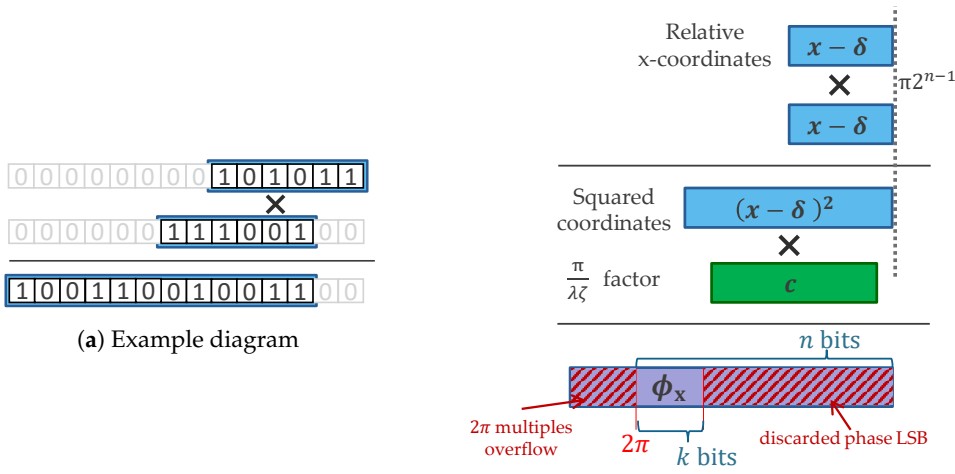

(**a**) Example diagram

(**b**) Fixed-point integer phase computation

**Figure 1.** Diagram of the fixed-point integer multiplications for computing the phase. (**a**) Fixed-point integer multiplication diagram with example bit patterns; the boxes indicate where significant bits are found (all other bits are assumed to be 0). (**b**) Proposed scheme for computing phase values. The bits encode the binary digits representing the negative powers of two, multiplied by $2\pi$. The first multiplication squares the relative coordinates and the second multiplies it again with the $c$ factors. The shaded red regions designate bits of $\phi_x$, which do not have to be computed, reducing computational requirements: the $(n - k)$ least significant bits (LSB), and the overflowing multiples of $2\pi$, which do not alter the result of the periodic trigonometric functions.

These low-precision $\phi_x$ and $\phi_y$ phase values are tightly packed together in registers, and all pairwise combinations are added together efficiently to obtain the phase $\phi = \phi_x + \phi_y$ in every pixel, as is detailed further in the "Implementation" section.

### 2.2. LUT-Free Low-Precision Approximations to the Complex Exponential

We would like find a way to compute cosine (and sine) functions efficiently, using fixed-point integers without the use of look-up tables (LUT). Accurately computing (co)sines takes significant resources, and has less utility in a low-precision computation environment. We are thus seeking approximations requiring minimal amount calculations. Since the appearance of holograms remains invariant under scaling, we should try to minimize the deviation from the true cosine only up to a constant scaling factor.

For the first approximation, we take a triangular signal with matching frequency and phase to a pure cosine. This is also known in the CGH literature as "Nishitsuji's approximation" [19]. This approximation consists of linear segments, described by an absolute difference operator. This can thus be computed easily, using only a difference and a conditional sign change. Within the interval of interest $[0, 2\pi[$, we obtain the following:

$$\cos(\phi) \approx TA(\phi) \propto |\phi - \pi| \qquad (6)$$

We call this function $TA$ the "triangular approximation". The error is small when the (co)sine is close to 0, given their Taylor expansion, but the deviation is quite significant near the peaks of the triangular wave. We aim to significantly reduce that error by making a modification. We first define the clipping function as follows:

$$\text{clip}(x, d) = \begin{cases} -d, & \text{if } x < d \\ x, & \text{if } |x| \leq d \\ d, & \text{if } x > d \end{cases} \qquad (7)$$

which saturates a signal between $-d$ and $+d$. This can be used to reduce the relative slope of the approximate function near the extremes of the cosine. We use the following:

$$\cos(\phi) \approx PA(\phi) \propto TA(\phi) + \text{clip}(TA(\phi), d) \qquad (8)$$

which we call the "piecewise approximation". It either doubles the computed $TA$ value, or adds a saturated version of the signal to itself when $|x| > d$. The relative slope is thus halved near the peaks and valleys of the cosine. We still need to determine the best value for $d$, so that the squared error $\|\cos(\phi) - PA(\phi)\|$ is minimal (using the Euclidean norm $\|\cdot\|$). This optimum can be found easily since the error function is convex, cf. Figure 2. The minimum is reached at $d = 0.7515$, where the error becomes $1.557 \times 10^{-3}$. This can be contrasted with the error of the triangular approximation, which equals to $2.272 \times 10^{-2}$ at optimal scaling, differing by an order of magnitude.

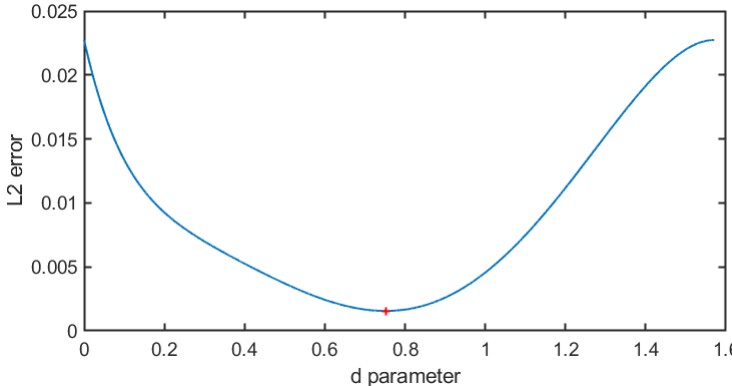

**Figure 2.** Euclidean norm of the error $\|\cos(\phi) - PA(\phi)\|$ as a function of $d$. The function is convex, reaching a minimum error of $1.557 \times 10^{-3}$ at $d = 0.7515$.

The resulting functions can be viewed in Figure 3. One can see that changes in the slopes of the smoothing of the piecewise approximation match the reference cosine much

better, reducing the error by a factor of $\approx 7$. This allows for the calculation of hologram wave fields with increased accuracy without sacrificing much extra calculation time. The sine function can be computed analogously by using a $\frac{\pi}{2}$ phase delay.

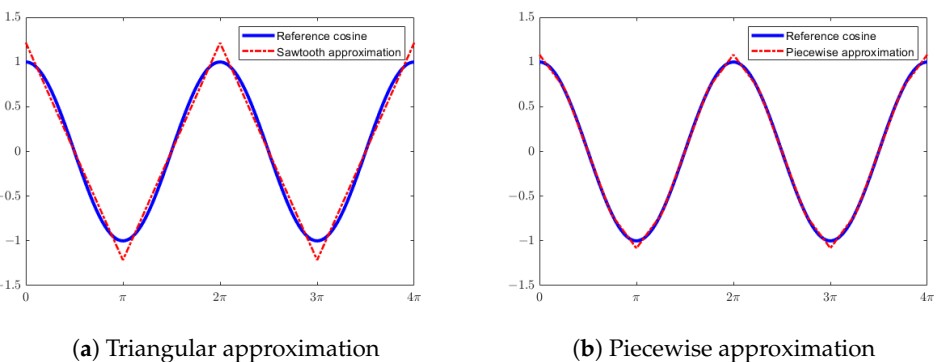

(**a**) Triangular approximation    (**b**) Piecewise approximation

**Figure 3.** Diagrams of the implemented LUT-free approximation functions. The shown approximations are scaled so that the squared error is minimal.

## 3. Implementation

This section details the GPU implementation, instructions and pseudo code. For the reference implementation, we allocate a single GPU thread per pixel, each looping over all points in the point cloud, numerically computing the expression (1) for every point. We utilize the intrinsic `__sincosf` function for computing the phase in the exponential function (since $\exp i\phi = \cos\phi + i\sin\phi$); this is significantly faster, but less precise, than the conventional single precision floating-point sine and cosine functions. We attempted to utilize a separable implementation for this floating-point version as well, but this resulted in a net increase in calculation time: the additional memory transfers outweigh the reduction in the amount of arithmetic operations. Therefore, a non-separated version of the PSF is used for the reference implementation.

The fixed-point versions of the algorithm operate in two phases. In the first phase, the separated packed phase bytes are computed for all $x$- and $y$-coordinates for all points. Since it is separable, computing a $M \times N$ resolution hologram with a point cloud consisting of $Q$ points needs $(M + N)Q$ bytes in total. When $Q$ becomes too large, the GPU memory may run out, so the calculations are performed in multiple point batches.

This first phase allocates one thread for every unique pixel coordinate and group of four points. Let us consider a square hologram, where $M = N$; since the GPU registers are 32-bit words, 4 bytes can be packed in a register. Therefore, every thread can compute an 8-bit phase value for $\phi_x$ (and $\phi_y$), packing them together for four points at a time in a single word (Figure 4). In total, we obtain $\frac{MQ}{4}$ threads, each computing two words of 4 phase bytes each: one for $\phi_x$ and one for $\phi_y$.

In the second phase, threads are grouped in 2D thread blocks (we used $16 \times 16$ threads per block), each assigned to a different pixel. The threads collectively load the relevant phase bytes in shared memory in small batches. For example, when there are $16 \times 16 = 256$ threads, each loading a single word (4 packed phase bytes), mapping to $16 + 16 = 32$ unique coordinates, every small batch processes $4 \times 256/32 = 32$ unique points per load. Then, every thread accesses the relevant packed phase bytes `pphase_x` and `pphase_y` in a short unrolled loop over all points in shared memory, cf. Figure 5.

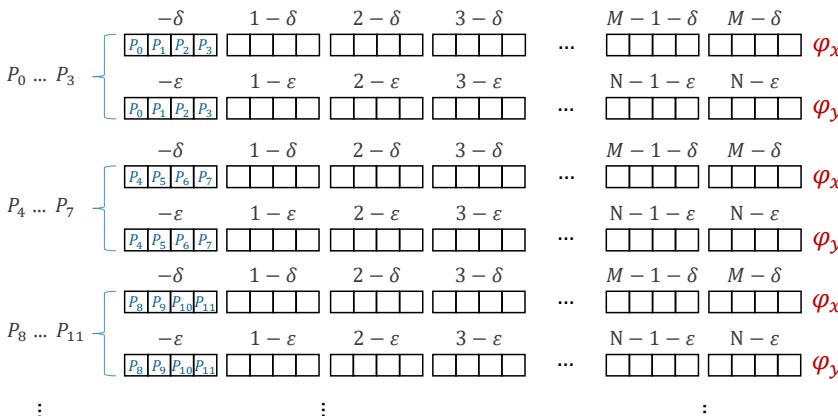

**Figure 4.** Memory layout of packed phase bytes for $\phi_x$ and $\phi_y$. Every 4 point phase gets packed together in a word, written to global GPU memory. In phase 2, the requisite combination of packed $\phi_x$ and $\phi_y$ can be efficiently processed together, using GPU SIMD instructions.

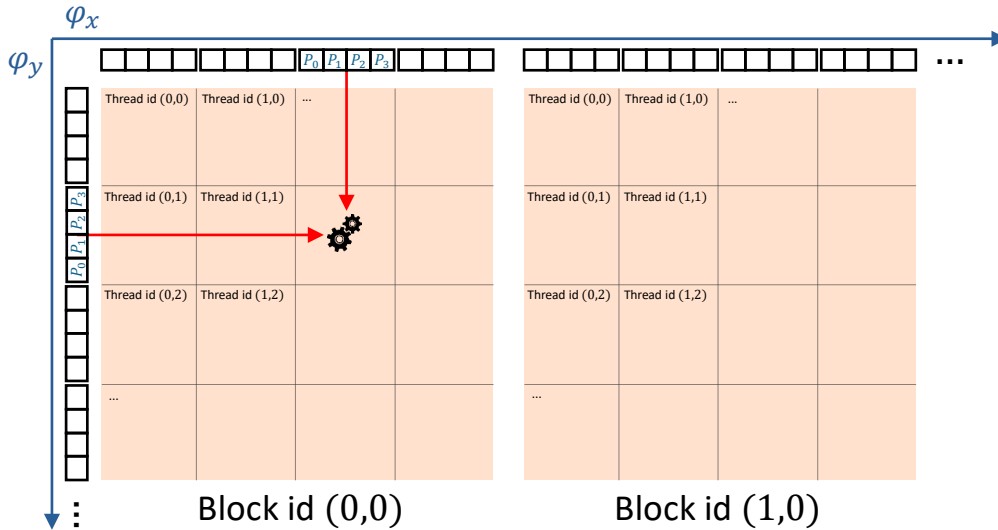

**Figure 5.** Simplified diagram of how hologram pixel values are computed. Every block loads the relevant packed phase bytes into shared memory, which can be combined together by every thread within the block depending on their thread coordinates. The resulting complex amplitude is accumulated within a register, whose final result is written to video memory as a hologram pixel value.

The next main instructions are as follows:

```
unsigned pphase = __vadd4(pphase_x, pphase_y);
unsigned cosval = __vabsdiffu4(pphase, 0x7F7F7F7F);
if constexpr (S_EXTENSION)
cosval += __vminu4(__vmaxu4(cosval, 0x22222222), 0x5E5E5E5E);
real_part += __dp4a(cosval, packed_amplitudes);
```

The double underscore prefix indicates the use of CUDA intrinsics, which are not standard C/C++. The `unsigned` are, by default, 32-bit integers. The `__vadd4` SIMD, intrinsic, performs a byte-wise addition, ignoring the overflow; this effect is desired, as overflows correspond to $2\pi$ multiples, which do not alter the outcome of the periodic sine and cosine functions. The triangular signal $TA(\phi)$ is computed with the byte-wise absolute difference instruction, `__vabsdiffu4` with `0x7F` = 127 per byte.

If the piecewise approximation extension flag `S_EXTENSION` is enabled, an extra set of instructions is executed. The "`if constexpr`" indicates that this is evaluated at compile

time. The optimal value for the clipping function in $PA(\phi)$, using 8 bits of precision, corresponds to $d = 30$. The byte-wise clipping happens, using the `__vmin4` and `__vmax4` instructions, with `64-d = 34 = 0x22` and `64+d = 94 = 0x5E`. This clipped signal is added to the original `cosval` to obtain a function proportional to the piecewise approximation function. Note that the conventional increment operator `+=` was deliberately chosen rather than another `__vadd4`; by design, we are guaranteed never to encounter byte-wise overflow here, so the faster regular increment is preferred. Finally, we used the relatively newer intrinsic instruction `__dp4a`, doing a byte-wise dot product: all bytes of the two input operators are multiplied pairwise and added together. The second operator `packed_amplitudes` contains the packed amplitudes of the corresponding 4 PSFs, thereby simultaneously computing and summing the real part of 4 PSFs to the accumulated pixel result "`real_part`".

The instructions are almost identical for the imaginary component of the signal, except for a phase delay since $\sin(x) = \cos(x - \frac{\pi}{2})$. Thus the sine can be obtained in the same manner by subtracting $256/4 = 64$ from all packed phase bytes with overflow (or equivalently by adding 192). The complete algorithm is summarized in Figure 6.

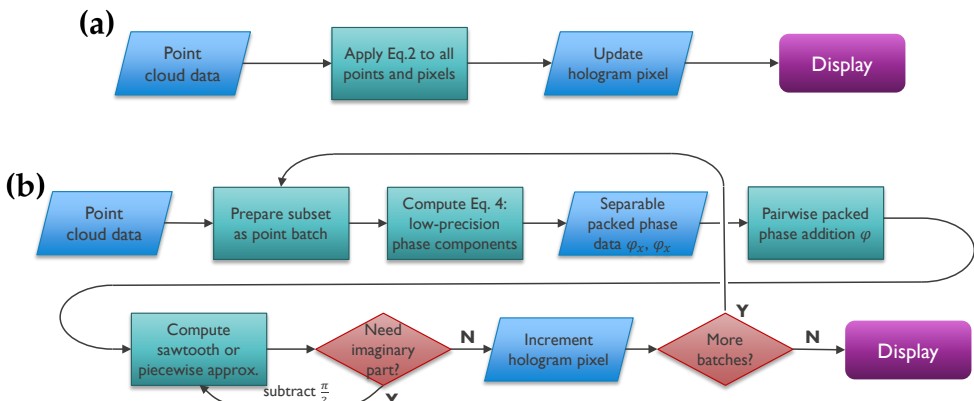

**Figure 6.** Algorithm flowcharts. (**a**) Reference floating-point point-cloud CGH algorithm. (**b**) Proposed fixed-point algorithm pipeline, illustrating the batch subdivision and the different calculation phases.

## 4. Experiments

This section consists of two sets of experiments: (1) generating holograms from images, so as to have an established baseline for the visual quality, and (2) holograms rendered from 3D point clouds to measure the differences in calculation speed. The algorithms were run on a machine with an AMD Ryzen Threadripper 3960X processor, 64 GB of RAM and a NVIDIA Geforce RTX 3080 GPU running a Windows 10 OS. They were implemented in C++17 with CUDA 11.2, enabling CUDA compute capability of 8.6 and using 32-bit floating-point precision for the reference implementation.

### 4.1. Visual Quality Experiments

We generated holograms with a resolution of $2048 \times 2048$ pixels, a pixel pitch of $p = 4$ µm and a wavelength of $\lambda = 532$ nm. The input data are a grayscale version of the standard "Peppers" test image, consisting of $512 \times 512$ pixels, decomposed into a point cloud (one point for every pixel). The virtual image plane is centered at the origin, displaced at a depth of 10 cm from the hologram plane.

The image quality is evaluated by first backpropagating the hologram with the angular spectrum method (ASM) [23] to the image plane, taking the magnitude and scaling and

quantizing it to 8 bits, and then evaluating the peak signal-to-noise ratio (PSNR), defined for 8-bit images as follows:

$$\text{PSNR}(I, \hat{I}) := 10 \log_{10} \left( \frac{255^2}{\|I - \hat{I}\|^2} \right) \tag{9}$$

where $\|\cdot\|$ is the Euclidean norm, $I$ is the reference image and $\hat{I}$ the approximation. The resulting images are shown on Figure 7. Comparing the quality of the reference floating-point algorithm to the triangular and piecewise fixed-point integer approximations, we obtain PSNR values of 34.58 dB and 51.19 dB, respectively.

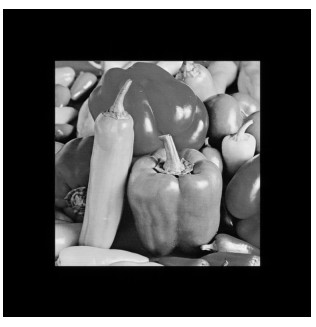 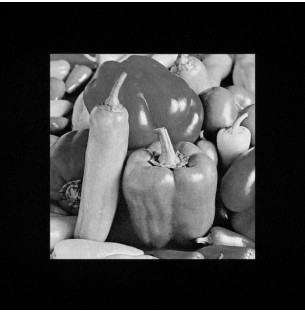 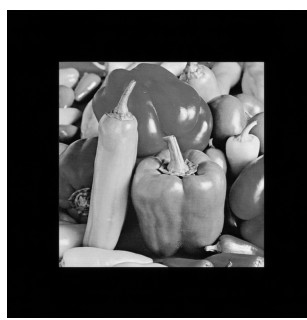

(**a**) Reference algorithm　　　(**b**) Triangular approximation　　　(**c**) Piecewise approximation

**Figure 7.** Reconstruction of the "peppers" image holograms, using ASM backpropagation and showing the hologram magnitude, computed using 3 different versions of the point-wise CGH algorithm implementations covered in this paper.

Please note that generally speaking, convolutional methods, such as discrete Fresnel diffraction or the ASM, should be used when creating holograms of planar images, which would be far more computationally efficient. However, these cannot directly calculate holograms of arbitrary 3D point clouds. Next, we investigate the impact of our proposed algorithm on 3D point cloud CGH.

### 4.2. Calculation Speed Experiments on a 3D Object

The 3D point cloud comes from a bi-plane model, consisting of $10^5$ points, each with their associated amplitudes (cf. Figure 8). The plane is centered laterally to match the hologram origin, and displaced to be 20 cm from the hologram plane. The dimensions of the plane point cloud along the main axes are $1.59 \times 0.39 \times 1.08$ cm.

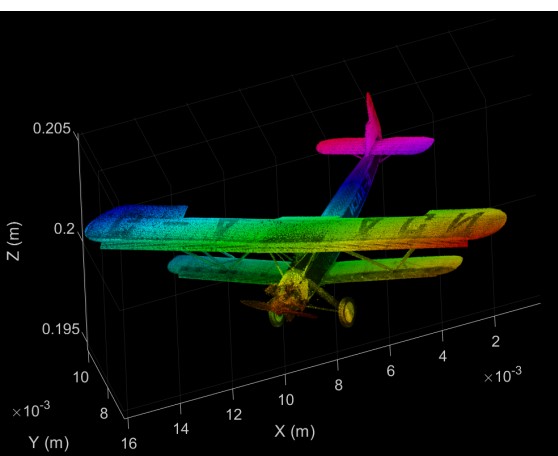

**Figure 8.** False color diagram of the "bi-plane" point cloud model, with its absolute coordinates. The intensity of the point color correspond to the actual amplitudes $a_j$ of their PSFs; the hue matches their relative $z$-position to improve visual interpretation.

The measured calculation times were averaged over 10 runs. The reference floating-point algorithm took 3330 ms to compute, the triangular approximation version took 1068 ms and the piecewise approximation version took 2594 ms. The reconstructions are shown on Figure 9. That means that the fixed-point integer versions are about $3.1\times$ and $1.3\times$ times faster, respectively.

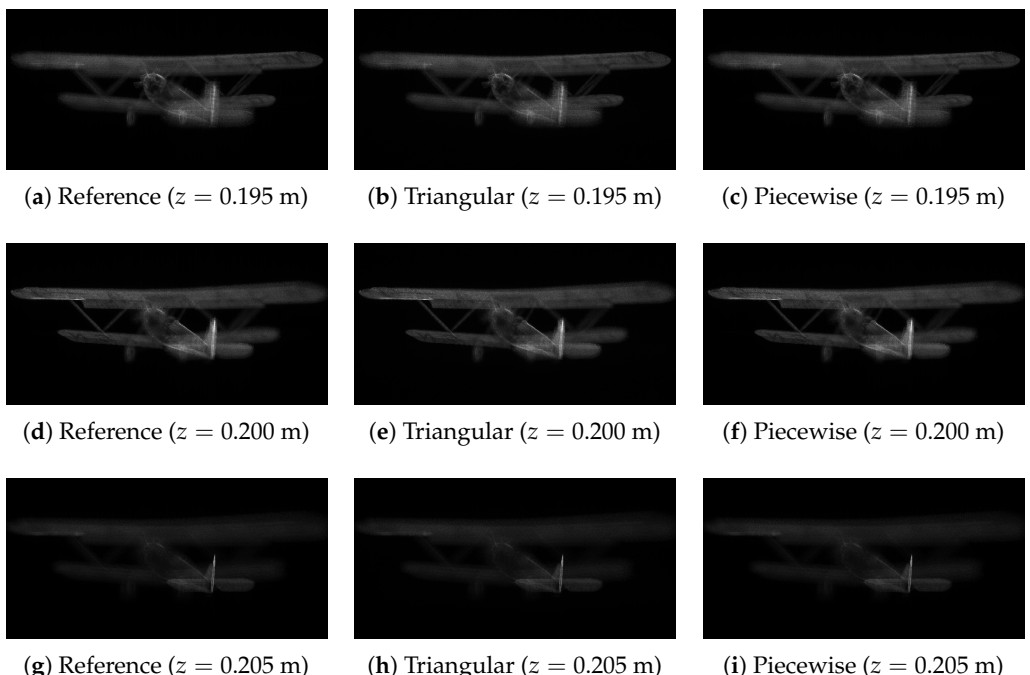

(**a**) Reference ($z = 0.195$ m)     (**b**) Triangular ($z = 0.195$ m)     (**c**) Piecewise ($z = 0.195$ m)

(**d**) Reference ($z = 0.200$ m)     (**e**) Triangular ($z = 0.200$ m)     (**f**) Piecewise ($z = 0.200$ m)

(**g**) Reference ($z = 0.205$ m)     (**h**) Triangular ($z = 0.205$ m)     (**i**) Piecewise ($z = 0.205$ m)

**Figure 9.** Various reconstructions of the "bi-plane" hologram, for the three algorithm versions, reconstructed at three different depths. The columns stand for the reference (**a**,**d**,**g**), triangular approximation (**b**,**e**,**h**) and piecewise approximation (**c**,**f**,**i**). The rows show the front (**a**–**c**), middle (**d**–**f**) and back (**g**–**i**) of the bi-plane in focus, respectively.

All results are summarized in Table 1.

**Table 1.** Summary of the results for the computed holograms comparing the different proposed algorithms. The best results in each column are indicated with boldface.

| Algorithm | PSNR (dB) | Calculation Time (ms) | Speedup Factor |
|---|---|---|---|
| Reference method | / | 3330 | 1.0 |
| Triangular approximation | 34.58 | **1068** | **3.1** |
| Piecewise approximation | **51.19** | 2594 | 1.3 |

## 5. Discussion

Fixed-point integer approximation algorithms can bring significant speed improvements to CGH applications. Because they are primarily used for visualization and because in holograms, the information is generally distributed over the entire hologram plane, there is much more tolerance for errors than for conventional natural image and video rendering, which can be leveraged for speed.

Many candidate CGH algorithms exist with potential for acceleration, but many tend to be rather complex, requiring several different transforms and operations, such as multiple fast Fourier transforms, non-uniform resampling, LUT use and non-linear occlusion operations. We opted to accelerate the point-cloud CGH algorithm, due to its simplicity, low memory requirements and high parallelizability, making it highly compatible for (simpler) FPGAs and ASICs.

The advantage of the CGH method's simplicity also has a drawback in its limited supported shading effects. Extensions of the method addressing the limitation bring increased complexity in the algorithms, making the use of specialized hardware systems more difficult. The main disadvantage of the proposed method is the quality loss w.r.t. reference point-cloud method, though this can be controlled by selecting the appropriate approximation and bit-width depending on the CGH application's needs. Moreover, it has to be noted that the phase packing phase takes a non-negligible time on a GPU, and SIMD instructions are noticeably slower than the conventional instructions in the current GPU models. The obtained relative speed ups may thus be increased further in the context of FPGA or ASIC systems, where there is more fine-grained control on the bit width of various variables, as well as broader possibilities for customized instructions and the reduction of power requirements.

In future work, we aim to implement optimized implementations on FPGA or ASIC systems, as well as refine the algorithm by supporting more visual effects and further reducing calculation requirements.

### 6. Conclusions

We presented a novel algorithm for efficiently computing point-cloud CGH, using low-precision fixed-point integers and LUT-free piecewise approximations to the sine and cosine functions, suitable for highly parallel computing systems. We report significant gains in calculation speed at high visual quality.

**Author Contributions:** Conceptualization, D.B.; methodology, D.B.; software, D.B.; validation, D.B. and P.S.; formal analysis, D.B.; investigation, D.B.; resources, D.B..; data curation, D.B. and P.S.; writing—original draft preparation, D.B.; writing—review and editing, P.S.; visualization, D.B.; supervision, D.B. and P.S.; project administration, D.B. and P.S.; funding acquisition, D.B. All authors have read and agreed to the published version of the manuscript.

**Funding:** This research was funded by Fonds Wetenschappelijk Onderzoek (12ZQ220N, VS07820N).

**Institutional Review Board Statement:** Not applicable.

**Informed Consent Statement:** Not applicable.

**Data Availability Statement:** The data supporting the findings of this study are available upon reasonable request from the corresponding author.

**Acknowledgments:** The Biplane point cloud model is courtesy of ScanLAB Projects.

**Conflicts of Interest:** The authors declare no conflict of interest.

### Abbreviations

The following abbreviations are used in this manuscript:

| | |
|---|---|
| 3D | Three-dimensional |
| ASIC | Application-specific integrated circuit |
| ASM | Angular spectrum method |
| CGH | Computer-generated holography |
| CUDA | Compute Unified Device Architecture |
| FPGA | Field-programmable gate array |
| GPU | Graphics processing unit |
| LSB | Least significant bit |
| LUT | Look-up table |
| SIMD | Single instruction, multiple data |
| PSNR | Peak signal-to-noise ratio |

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
