# Peer review of "Fast Low-Precision Computer-Generated Holography on GPU"

_applsci, doi:10.3390/app11136235_

Round 1

Reviewer 1 Report

Please discuss the Sawtooth and Piecewise approximation used in the paper, more detail required. And why is the PSF approach chosen over Ray-Based or the Polygon ... discuss. 

Author Response

We would like to thank the reviewer for the positive evaluation of the manuscript and for the comments and suggestions.

A) Following the suggestion of the reviewer, more explanation and discussions were added to section 2.2, “LUT-free low-precision approximations to the complex exponential”. We renamed the “sawtooth” to “triangular”, as this definition is more accurate when describing waveforms. In summary, we’ve detailed the rationale for seeking approximate functions, expanded the explanation on why the triangular wave is simpler to calculate and explained more in detail why the triangular approximation is inaccurate at the peaks and how to mitigate this with the newly introduced piecewise approximation; we also added a short discussion at the end of the section.

B) The PSF approach is a subset of ray-based approaches, where the point-spread function is computed for all hologram pixels. As detailed in the introduction, the reason for our choice is the relatively simple mathematical expression of PSFs and the inherent parallelizability of the linear superpositions.

Other ray-based approaches such as holographic stereograms can also be accelerated, but generally introduce more significant approximations to the wavefield, leading to i.a. loss of parallax and limited viewing depth ranges.

Polygonal CGH algorithms are very accurate, but are algorithmically much more complex, requiring many FFTs, frequency-domain resampling and occlusion operators, making them less suitable for implementation in FPGAs and ASICs.

We added a discussion section in the manuscript before the conclusion, detailing among other things the reasoning for choosing the point-cloud PSF CGH algorithm.

Reviewer 2 Report

In general, two classes are mentioned for CGH algorithms: point-based method (PBM) and a polygon-based method(PolyBM). It is demonstrated that the polygon-based method reduces a vast amount of sampling units compared with the point-based method. 

The paper proposed an optimization of the PBM method. The mathematical expression of PSF is relative simple and represents a good candidate for parallel implementation on GPU. The goal and contributions are presented very clear in the first section (Introduction). In the paper is mentioned the paper " Review of fast methods for point-based, the author is cited for his contribution for the paper " Computer-generated holograms by multiple wavefront recording plane method with occlusion culling"". 

I would recommend the following: - section 3: flowchart or pseudocode can be useful to highlight valuable and detailed contributions that are presented as collages and the logic of the paper can be much easier to follow; - section 2: the diagrams in figure 3 can be presented in the same line (not complicated); - section 4: the experiments are well conducted. Comparison tables with results are welcome. section 5: the advantages and disadvantages of the method need to be mentioned (self-evaluation)

Author Response

We would like to thank the reviewer for the positive evaluation of the manuscript and for the detailed comments and suggestions.

A) Following the reviewer’s suggestion, a new figure (#6) with the flowchart of the complete algorithm has been added for better clarity.

B) The diagrams of Figure 3 have been rescaled and are now presented on the same line.

C) We included a table at the end of section 4 summarizing and comparing the experimental results.

D) Following the reviewer’s comment, we included a new discussion section before the conclusion, discussing among other things the various advantages and disadvantages of the proposed algorithm.